# Activation by cleavage of the epithelial Na⁺ channel α and γ subunits independently coevolved with the vertebrate terrestrial migration

Xue-Ping Wang[1], Deidra M Balchak[1], Clayton Gentilcore[1], Nathan L Clark[2], Ossama B Kashlan[1,3]*

[1]Department of Medicine, University of Pittsburgh, Pittsburgh, United States; [2]Department of Human Genetics, University of Utah, Salt Lake City, United States; [3]Computational and Systems Biology, University of Pittsburgh, Pittsburgh, United States

**Abstract** Vertebrates evolved mechanisms for sodium conservation and gas exchange in conjunction with migration from aquatic to terrestrial habitats. Epithelial Na⁺ channel (ENaC) function is critical to systems responsible for extracellular fluid homeostasis and gas exchange. ENaC is activated by cleavage at multiple specific extracellular polybasic sites, releasing inhibitory tracts from the channel's α and γ subunits. We found that proximal and distal polybasic tracts in ENaC subunits coevolved, consistent with the dual cleavage requirement for activation observed in mammals. Polybasic tract pairs evolved with the terrestrial migration and the appearance of lungs, coincident with the ENaC activator aldosterone, and appeared independently in the α and γ subunits. In summary, sites within ENaC for protease activation developed in vertebrates when renal Na⁺ conservation and alveolar gas exchange were required for terrestrial survival.

*For correspondence:
obk2@pitt.edu

**Competing interest:** The authors declare that no competing interests exist.

## Editor's evaluation

The authors show that enzymatic regulation of a sodium-permeable channel that aids in gas exchange and fluid homeostasis likely evolved at the water to land transition in vertebrates. This work is impactful as it details a critical period in the evolution of terrestrial vertebrates.

## Introduction

The migration of vertebrates to a terrestrial habitat began ~380 million years ago and was driven by competitive pressure and the relative availability of resources (*Long and Gordon, 2004*). Survival in a terrestrial environment required several adaptations, including those for respiration, extracellular fluid volume balance, and osmoregulation. Aldosterone signaling has a significant role in extracellular fluid homeostasis for tetrapods and depends on three proteins that appeared during the evolution of vertebrates: aldosterone synthase, the mineralocorticoid receptor (MR), and 11β-hydroxysteroid dehydrogenase (11β-HSD2) (*Rossier et al., 2015*). One of the major endpoints of aldosterone signaling through MR is increased activity of the epithelial Na⁺ channel (ENaC) in the kidney, which results in enhanced Na⁺ and fluid retention. Having originally evolved in a marine environment, these mechanisms to achieve electrolyte homeostasis may have required adaptive modifications as the ancestors of today's terrestrial vertebrates moved to a relatively dry environment.

**Figure 1.** Sequence conservation in epithelial Na⁺ channel (ENaC) subunits. (**A**) Space filling model of ENaC (pdb code: 6BQN) with plane indicating position of outer membrane border. The α and γ subunits are white and gray, respectively. The β subunit is colored by domain as indicated in panel B. Intracellular structures are absent in this structural model. (**B**) Linear model of human ENaC subunits showing domain organization and highlighting position of polybasic cleavage sites and PY motifs. (**C**) Sequences (*Supplementary file 1*) were aligned using MUSCLE (*Edgar, 2004*; *Figure 1—source data 1*). Residue symbol sizes are proportional to frequency at a given position. Key features in the sequence are indicated, as are the approximate position of α helices (rounded rectangles) and β-strands (arrows). Colors correspond to protein domains, as indicated in panel B. The GRIP (gating release of inhibition by proteolysis) domain is unique to ENaC subunits in the ENaC/Deg family. The P1 and P2 β-strands in the GRIP domain are not indicated, but are likely near the inhibitory tract and distal site, respectively, if present.

The online version of this article includes the following source data for figure 1:

**Source data 1.** Multiple sequence alignment of proteins in *Supplementary file 1*.

ENaCs are heterotrimers comprising α, β, and γ subunits that each have intracellular amino and carboxyl termini, two transmembrane domains, and a large extracellular region (*Figure 1A, B*). An ENaC δ subunit can substitute for the α subunit in assembled channels, resulting in channels with distinct functional properties (*Giraldez et al., 2012*). However, the δ subunit's absence in mice and rats has led to fewer reports of its function and physiology, as most of this work has been performed in these model systems. ENaC subunits belong to the ENaC/Degenerin family of proteins that form trimeric cation-selective channels with similar extracellular folds (*Jasti et al., 2007*; *Noreng et al., 2018*; *Kashlan et al., 2011*). Aldosterone regulation of ENaC activity and abundance depends on

essential sequences in both the extracellular and intracellular domains. These include polybasic tracts in extracellular GRIP (gating release of inhibition by proteolysis) domains unique to ENaC subunits; however, the polybasic tracts of ENaC are not present in all vertebrate lineages (*Figure 1B*; *Noreng et al., 2018*; *Hughey et al., 2004a*; *Bruns et al., 2007*; *Noreng et al., 2020*). Aldosterone promotes double cleavage of the α and γ subunits at polybasic tracts, which liberates embedded inhibitory tracts and converts dormant channels to constitutively open ones (*Kleyman et al., 2018*; *Frindt and Palmer, 2009*; *Frindt and Palmer, 2015*; *Terker et al., 2016*), thereby leading to increased fluid retention. ENaC subunits also feature PY motifs on their intracellular C-termini that are important for protein degradation (*Figure 1B*; *Snyder et al., 1995*; *Schild et al., 1996*). These motifs recruit WW-domain containing Nedd4-2, whose binding results in the ubiquitylation of ENaC subunit N-termini and enhanced endocytosis and degradation (*Staub et al., 1996*; *Rizzo and Staub, 2015*). Nedd4-2-dependent reduction of ENaC abundance is inhibited by aldosterone (*Rizzo and Staub, 2015*). These complementary regulatory mechanisms help aldosterone minimize Na⁺ wasting and fluid loss during volume contraction.

Here, we investigated the evolution of ENaC regulatory mechanisms to determine which features coevolved with the marine–terrestrial transition. We consistently found both activating cleavage sites in the ENaC α and γ subunits of terrestrial vertebrates, while they appeared only sporadically in fishes. We confirmed that cleavage occurred at sites found in the γ subunit from Australian lungfish, leading to channel activation. Phylogenetic analysis and likelihood ratio tests showed a coevolutionary dependence of the polybasic tracts with each other. They also showed a coevolutionary dependence of tandem polybasic tracts with terrestrial status and with lungs. Analysis of ancestral reconstructions strongly suggests that the polybasic tracts appeared independently in the α and γ subunits. Similar analyses of the PY motif showed no coevolutionary pattern and that the PY motif first arose in an ancient ancestral ENaC subunit. Our data suggest that changes associated with adaptation to terrestrial life provided selective pressure for the development of ENaC activation by cleavage.

## Results
### ENaC subunit sequence conservation and evolution

The subphylum Vertebrata encompasses the vast majority of species in the phylum Chordata. We identified ENaC subunit sequences in each of the classes of Vertebrata, including all four ENaC subunits in each mammalian order. In order Rodentia, we found the δ subunit in Spalacidae (blind mole rats; accession XP_008850903), but in neither Muridae (mice and rats) nor Cricetidae (hamsters and voles), suggesting that the δ subunit was lost on the lineage to mice and rats after the divergence of Spalacidae ~34 million years ago (*Blanga-Kanfi et al., 2009*; *Steppan, 2017*). Notably, although we found ENaC subunits in Sarcopterygii (lobe-finned fishes) and the ray-finned Polypteriformes (ropefish), we did not find ENaC subunits in any other ray-finned fishes. Instead, BLAST searches using human ENaC subunits identified ENaC-related acid-sensing ion channels (ASICs) and uncharacterized ENaC-like proteins. This suggests ray-finned fishes lost all ENaC subunits after the early divergence of the Polypteriformes order and before the divergence of Chondrostei (sturgeons) and Neopterygii (gars and teleosts). Similarly, we only found ENaC-related proteins in nonvertebrate chordates (e.g., tunicates or cephalochordates).

To examine sequence conservation within ENaC sequences, we aligned ENaC subunit sequences found in each of the classes and top teleost and nonvertebrate chordate hits identified in a BLAST search using human ENaC subunits (*Supplementary file 1*). Sequence conservation was highest in the transmembrane helices (TM1 and TM2) and defined extracellular secondary structures in the ENaC and ASIC1 structures (*Figure 1C* and *Figure 1—source data 1*; *Jasti et al., 2007*; *Noreng et al., 2018*). Sequence conservation was lowest in the intracellular N- and C-termini, extracellular loops, and the GRIP domain unique to ENaC subunits. Sequences associated with key functions demonstrated varied conservation. For example, G/S-X-S in TM2 contributing to ion selectivity (*Kashlan and Kleyman, 2011*) is well conserved within a conserved region (*Figure 1C*). The PY motif in the intracellular C-terminus is also well conserved, but in a poorly conserved region. The region in the GRIP domain containing the embedded inhibitory tracts and cleavage sites is poorly conserved.

To examine the evolution of ENaC subunits, we assessed the phylogenetic relationship of our 53 assembled sequences with maximum likelihood methods (*Figure 2*). In the resulting tree, we observed

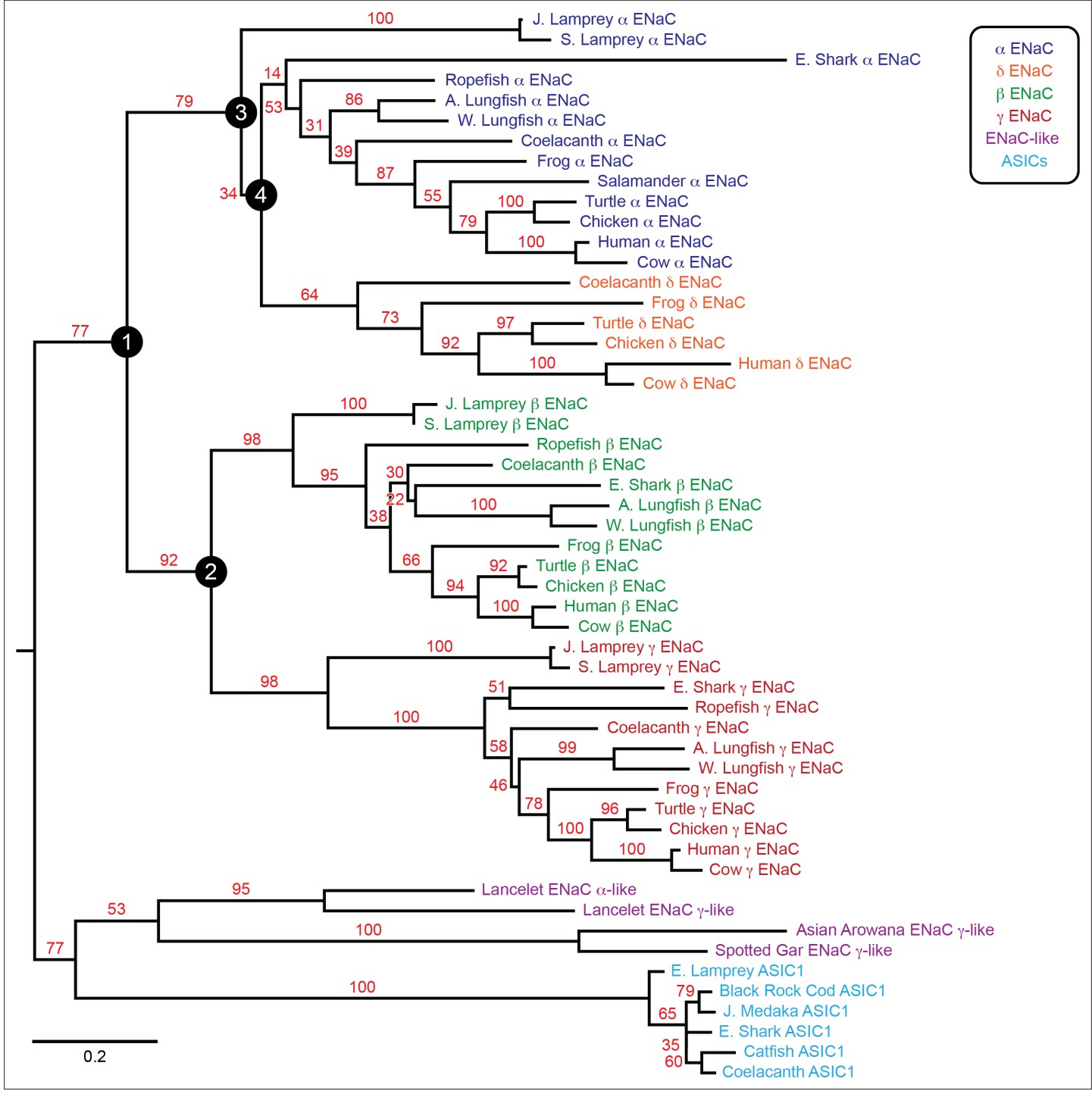

**Figure 2.** Phylogenetic tree of epithelial Na+ channel (ENaC) subunits. Maximum likelihood tree calculated from ENaC subunit sequences of marine species and select terrestrial vertebrates, and ENaC-related proteins. Branch support bootstrap values are shown. Scale bar indicates the number of substitutions per site. Key ancestral nodes are indicated by circled numbers. A. Lungfish = Australian Lungfish, E. Shark = Elephant Shark, E. Lamprey = European River Lamprey, J. Lamprey = Japanese Lamprey, J. Medaka = Japanese Medaka, S. Lamprey = Sea Lamprey, W. Lungfish = West African Lungfish.

six clades: ENaC α, β, γ, and δ subunits, ASICs, and ENaC-like proteins. Each ENaC subunit clade largely recapitulated the expected phylogenetic relationships, with jawless fishes (lampreys) rooted closest to the base of the clade, followed by cartilaginous fishes (elephant shark), lobe-finned fishes (coelacanth and lungfishes), reptiles and birds, and mammals. Our evolutionary model suggests three

**Table 1.** Polybasic sequences aligning with human epithelial Na$^+$ channel (ENaC) subunit proximal (site 1) and distal (site 2) cleavage sites.

Terr., terrestrial. Dashes indicate the absence of polybasic tract. [1]The ENaC δ subunit in mammals has a polybasic sequence in the aligned region, but is not cleaved in human channels (*Haerteis et al., 2009*). Ø, no sequence was available. (···), GRIP (gating release of inhibition by proteolysis) domain sequence was missing in the available sequence. Species abbreviations are as in *Figure 2*.

| Animal | Terr. | Lungs | α site 1 | α site 2 | δ site 1 | δ site 2 | β site 1 | β site 2 | γ site 1 | γ site 2 |
|---|---|---|---|---|---|---|---|---|---|---|
| S.Lamprey | | | — | — | Ø | Ø | — | — | — | — |
| J.Lamprey | | | — | — | Ø | Ø | — | — | — | — |
| E.Shark | | | (···) | (···) | Ø | Ø | — | — | — | RQHR |
| Ropefish | | X | — | — | Ø | Ø | — | — | RKRR | NRKR |
| Coelacanth | | | RSNR | — | — | — | KRER | — | — | VKQR |
| W.Lungfish | | X | — | — | Ø | Ø | — | — | — | — |
| A.Lungfish | | X | — | — | Ø | Ø | — | — | RKLR | RQYR |
| Frog | X | X | RVKR | RVSR | — | — | — | — | RSKR | KRTR |
| Salamander | X | X | RERR | RVRR | Ø | Ø | Ø | Ø | Ø | Ø |
| Turtle | X | X | RSPR | RHKR | — | — | — | — | KVRR | NKRK |
| Chicken | X | X | RTSR | RQKR | — | — | — | — | KVRR | RKRK |
| Cow | X | X | RSRR | RGVR | — | *RLQR*[1] | — | — | RKRR | RKRK |
| Human | X | X | RSRR | RRAR | — | *RLQR*[1] | — | — | RKRR | RKRK |

gene duplication events for ENaC subunits, as previously reported (*Studer et al., 2011*). The first duplication (node 1) gave ancestral β/γ subunits (node 2) and α/δ subunits (node 3). The second duplication (node 2) gave ancestral β and γ subunits. Both of these duplication events occurred in ancestors of jawless fishes at least 550 million years ago. The third duplication (node 4) leading to the divergence of α and δ subunits had more associated uncertainty. Given the presence of α and δ subunits in coelacanth, the duplication likely occurred before the divergence of the coelacanth and tetrapods.

## Polybasic tracts in the ENaC GRIP domains varied over time

We then examined the sequence conservation of the polybasic tracts in the ENaC subunit GRIP domains required for ENaC activation by cleavage. In mammals, the GRIP domains of the α and γ subunits are each subject to double cleavage, leading to the release of embedded inhibitory tracts and channel activation. The β and δ subunits are not similarly processed. The proprotein convertase furin cleaves both the proximal and distal sites of the α subunit, and the proximal site of the γ subunit (*Hughey et al., 2004a*). Cleavage distal to the γ subunit inhibitory tract can be catalyzed by several proteases at the cell surface, including prostasin at a polybasic tract (*Carattino et al., 2008a*; *Balchak et al., 2018*). Similar results were reported for ENaC from *Xenopus laevis* (*Wichmann et al., 2018*). We inspected our multiple sequence alignment for polybasic tracts that aligned closely with the tracts in mammalian and frog α and γ subunits (*Figure 1—source data 1*). We found polybasic tracts aligning with the proximal and distal sites of the human α and γ subunits for all tetrapod α and γ subunits (*Table 1*). We also found both sites present in the γ subunit from *Erpetoichthys calabaricus* (Ropefish) and *Neoceratodus forsteri* (Australian lungfish), but not in *Protopterus annectens* (West African lungfish). In coelacanth, we identified single polybasic tracts in the α, β, and γ subunits. The elephant shark's γ subunit also had a single distal polybasic tract. The human δ subunit sequence also exhibits a polybasic tract in this region, but experimental evidence shows that it is not cleaved (*Haerteis et al., 2009*).

## Australian lungfish ENaC γ subunit is cleaved at GRIP domain polybasic tracts

To determine whether apparent cleavage sites are functional in a subunit rooted before the emergence of tetrapods, we examined the ENaC γ subunit from Australian lungfish (Aγ) in *X. laevis*

oocytes. Aγ has two furin cleavage motifs predicted to lead to activation (*Table 1*, *Figure 3A*). To isolate functional effects to Aγ, we coexpressed an α subunit from mouse lacking residues excised by furin (mouse αΔ206–231; mα$^Δ$), rendering it incapable of proteolytic activation (*Carattino et al., 2008b*). We also coexpressed a mouse β subunit truncated before the C-terminal PY motif (mouse βR564X; mβ$^T$) to decrease channel turnover (*Shimkets et al., 1997*). All γ subunits contained a C-terminal hemagglutinin (HA) epitope tag to facilitate detection (*Figure 3A*). Oocytes expressing ENaC were conjugated with a membrane impermeant biotin reagent to label surface proteins. We then lysed the oocytes, isolated biotin-labeled proteins using NeutrAvidin beads, and analyzed whole cell and surface enriched samples by western blot using anti-HA antibodies. We detected full-length and cleaved forms of mγ and Aγ (*Figure 3B*). In both cases, the proportion of cleaved γ subunit was higher in the surface pool than the total pool (*Figure 3C*), consistent with trafficking-dependent processing reported for mammalian ENaC (*Hughey et al., 2004b*). To confirm that cleavage occurred at the predicted sites, we mutated the terminal Arg in both sites to Ala (Aγ$^{2A}$). When we expressed Aγ$^{2A}$ in oocytes, the higher molecular weight band remained readily apparent while the lower molecular weight band largely disappeared. Comparison of quantified band densities confirmed that mutation of predicted cleavage sites in Aγ greatly diminished apparent cleavage (p = 0.007). The extent of any cleavage was similar in total and surface pools for Aγ$^{2A}$, in contrast to Aγ.

When we measured whole cell currents in oocytes expressing Aγ, we found that mutating predicted furin sites decreased ENaC-mediated currents by 82% (p < 0.0001), consistent with mutation precluding proteolytic activation (*Figure 3G, H*). Notably, currents from oocytes expressing Aγ$^{2A}$ were similar to currents from oocytes lacking γ subunits altogether. Similar surface expression levels for Aγ and Aγ$^{2A}$ (*Figure 3D–F*) suggest that differences in expression or surface delivery do not account for the differences we observed in ENaC-mediated currents. Together, these data provide evidence that Aγ undergoes activating cleavage at the predicted furin cleavage sites.

## GRIP domain cleavage sites coevolved with each other

Activation by cleavage requires two cleavage events within a single subunit and the release of embedded inhibitory tracts (*Bruns et al., 2007*; *Sheng et al., 2006*). The data in *Table 1* suggest that sites 1 and 2 coevolved with each other. To test this idea, we calculated the probability that the sites coevolved with each other using nested likelihood models based on our model of ENaC evolution in BayesTraits (see *Supplementary file 2*; *Meade, 2022* ; *Pagel et al., 2004*). Each sequence was assigned the traits of the presence or absence of 'site 1' and 'site 2', as indicated in *Table 1*. All models considered the evolutionary gains and losses of these traits over the full phylogenetic tree capturing the four ENaC subunits (*Figure 2*).

To test for dependence between the two sites, we contrasted two nested models, an independent model and a dependent model. The independent model contained two parameters: an appearance rate for both sites and a loss rate for both sites. The loss or gain of a site did not depend on the status of the other site, and therefore served as the null hypothesis. The dependent model added two free parameters by having separate gain and loss rates depending on the status of the other site. A likelihood ratio test between nested models showed a strong preference for the dependent model (p = 0.01). Fits to the dependent model suggest little selection pressure in the change from 0 sites to one site, with loss rates 11-fold higher than gain rates. In contrast, in the change from one site to two sites, gain rates were sevenfold higher than loss rates. This is congruent with the requirement for two cleavage sites within a single subunit's GRIP domain for channel activation, and supports the notion that selection pressure derives from the functional consequence of double cleavage (*Bruns et al., 2007*; *Sheng et al., 2006*).

## GRIP domain double cleavage coevolved with the terrestrial migration and with Lungs

The data in *Table 1* also suggest that the terrestrial migration or the development of lungs may have provided the selection pressure for channel activation by cleavage. ENaC function is also important in aldosterone-insensitive tissues, including the airway, where control of airway surface liquids is essential for lung function. We first calculated the likelihood of site 1, site 2, or both sites coevolving with the terrestrial transition using nested likelihood models. Each sequence was assigned the traits of the presence or absence of 'site 1', 'site 2', or 'tandem sites', and 'terrestrial' or 'marine' states, as

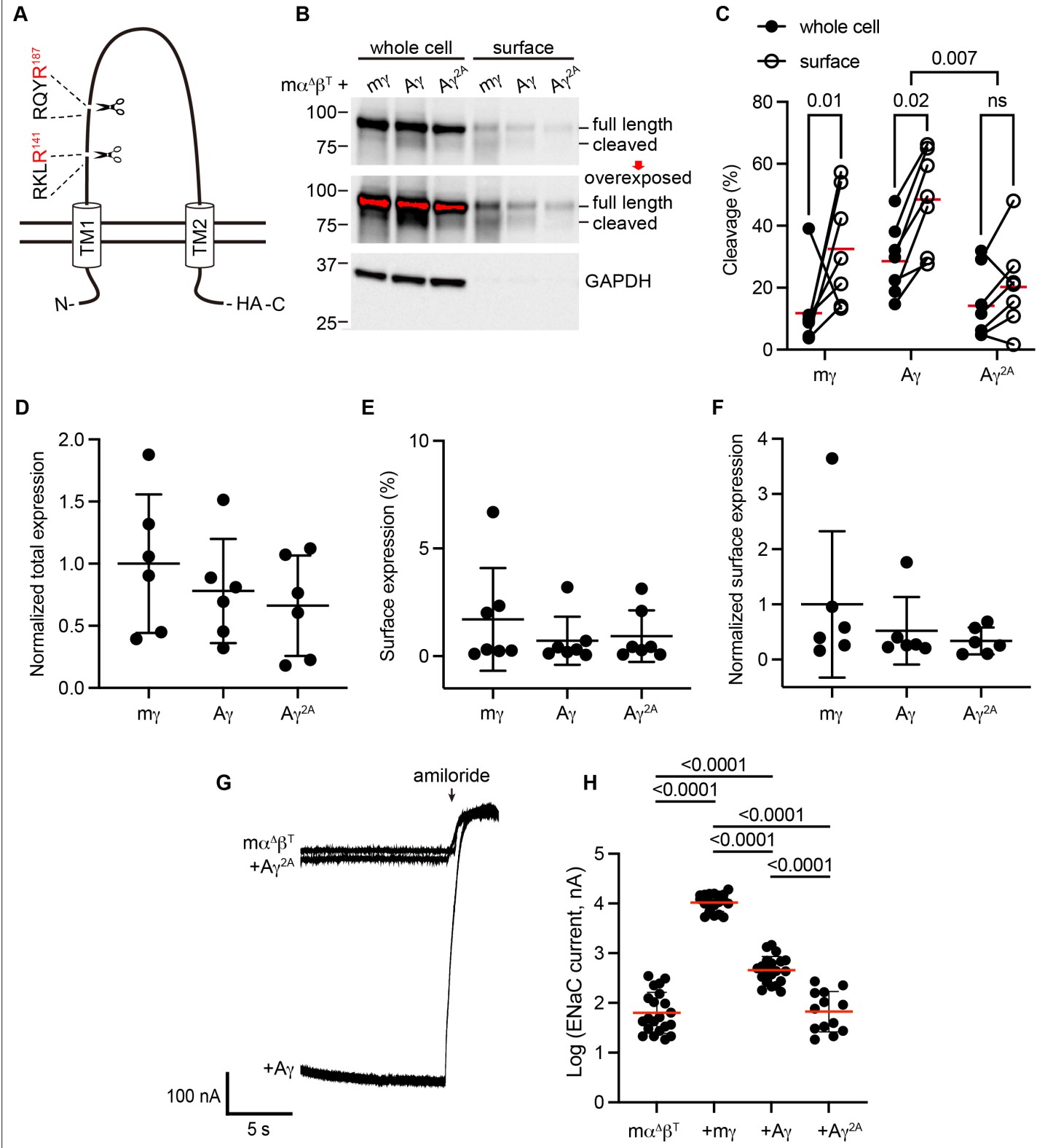

**Figure 3.** Predicted cleavage sites in the epithelial Na⁺ channel (ENaC) γ subunit from Australian lungfish are functional. (**A**) Schematic of Aγ topology. Aγ has two predicted furin cleavage sites in its extracellular GRIP (gating release of inhibition by proteolysis) domain. All γ subunits were labeled with C-terminal epitope tags to facilitate detection of full-length subunits and the larger of the cleaved fragments. (**B**) *Xenopus* oocytes were injected with cRNAs encoding mα$^\Delta$, mβ$^T$, and hemagglutinin (HA)-tagged γ subunits, as indicated. One day after injection, whole cell lysates and cell surface isolates

*Figure 3 continued on next page*

*Figure 3 continued*

were blotted and probed for HA and glyceraldehyde-3-phosphate dehydrogenase (GAPDH). Full-length and cleaved bands are indicated and band densities were quantified. An overexposed blot is shown to highlight cell surface bands. Over exposed areas are red. (**C**) Cleavage %, calculated as cleaved/(cleaved + full length) × 100 is shown. Data were analyzed by repeated measures two-way analysis of variance (ANOVA) with Šidák's multiple comparison test. p values are shown for indicated comparisons. Cleavage was also greater for Aγ than for mγ (p = 0.05). (**D**) Normalized total expression was calculated by normalizing the sum of full-length and cleaved bands to the mean of mγ after normalizing each sample for loading based on GAPDH from the same blot. (**E**) Surface expression % was calculated using band densities adjusted for the fraction of the respective sample loaded. (**F**) Normalized surface expression was calculated by multiplying values from the same sample in D and E, and then normalizing to the mean of mγ. Data in D–F were analyzed by one-way ANOVA with Tukey's multiple comparison test. No significant differences between groups were found. Note that due to the lack of GAPDH data for one blot, the number of replicates for D and F (*n* = 6) is one fewer than for C and E (*n* = 7). (**G**) Whole cell currents were measured in injected oocytes by two-electrode voltage clamp, with voltage clamped at −100 mV. Representative traces of indicated subunit combinations are shown. Currents were continuously recorded in a bath solution containing 110 mM Na⁺. The ENaC-blocking drug amiloride (100 μM) was added at the end of each experiment to determine the ENaC-mediated current. (**H**) Log-transformed amiloride-sensitive inward currents are plotted, and were analyzed by one-way ANOVA followed by Tukey's multiple comparison test. p values for comparisons where p < 0.05 are shown. Bars indicate mean values; errors shown are standard deviation (SD).

The online version of this article includes the following source data for figure 3:

**Source data 1.** All uncropped blots used for quantification are shown with full-length and cleaved bands indicated.

indicated in *Table 1*. The independent model contained three parameters: a site appearance and loss rate, and a rate for transition from marine to terrestrial status (see *Supplementary file 2*). The independent model does not allow a relationship between rates of site gain or loss with respect to marine or terrestrial state, and therefore served as the null hypothesis. The dependent model added two free parameters by having separate site gain and loss rates in the marine and terrestrial states. A likelihood ratio test between nested models showed a strong preference for the dependent model for site 1 (p = 0.0053) and for tandem sites (p = 0.0098), but not for site 2 (p = 0.26). Fits to the dependent model suggest that in the marine state, site appearance and disappearance were dynamic, with a disappearance rate being fivefold higher than the appearance rate. However, once in the terrestrial state, both rates dropped to 0, supporting the notion of selection pressure favoring the presence of cleavage sites in the terrestrial state. We then calculated the likelihood of site 1, site 2, or both sites coevolving with the development of lungs using analogous procedures. A likelihood ratio test between nested models showed a preference for the dependent model for site 1 (p = 0.031) and for tandem sites (p = 0.018), but not for site 2 (p = 0.94).

## ENaC expression in fishes with lungs and amphibians

Lungs coevolved with the terrestrial migration of vertebrates (p = 0.0004 in our dataset), and may have driven the coevolution of GRIP domain polybasic tracts with lungs that we observed. However, ENaC subunit transcripts were not detected in lung tissues from either of the lungfish species examined (*Uchiyama et al., 2015*; *Uchiyama et al., 2012*). To further investigate the role of ENaC in fishes with lungs, we examined the tissue distribution of ENaC transcripts in ropefish, which evolved lungs independently from tetrapods (*Graham and Wegner, 2010*). We observed clear bands for the transcripts of all three ENaC subunits in the gills and kidneys of the ropefish (*Figure 4A*), which are important organs for ion homeostasis. In the lung, we observed a clear band for the β subunit, a faint band for the α subunit, and a barely perceptible band for the γ subunit. In the other tissues examined, we observed broad expression of β subunit transcripts, faint bands for the α subunit in the liver and intestine, and no bands for the γ subunit. Taken together with the lungfish data (*Uchiyama et al., 2015*; *Uchiyama et al., 2012*), these data suggest that transcripts for ENaC subunits with GRIP domain polybasic tracts (*Table 1*) are readily detected at important sites of ion exchange, but are difficult to detect in lungs.

We also investigated the tissue distribution of ENaC subunit transcripts from the African clawed frog, in which both the α and γ subunits have GRIP domain cleavage sites (*Wichmann et al., 2018*). Similar to mammals, we detected bands for α, β, and γ subunit transcripts in both kidneys and lungs (*Figure 4B*). In contrast to humans where bands for the δ subunit were relatively faint for both kidneys and lungs (*Waldmann et al., 1995*), we observed a strong band for the δ subunit transcript in the kidney, and no band in the lung. The expression pattern we observed is consistent with a previous

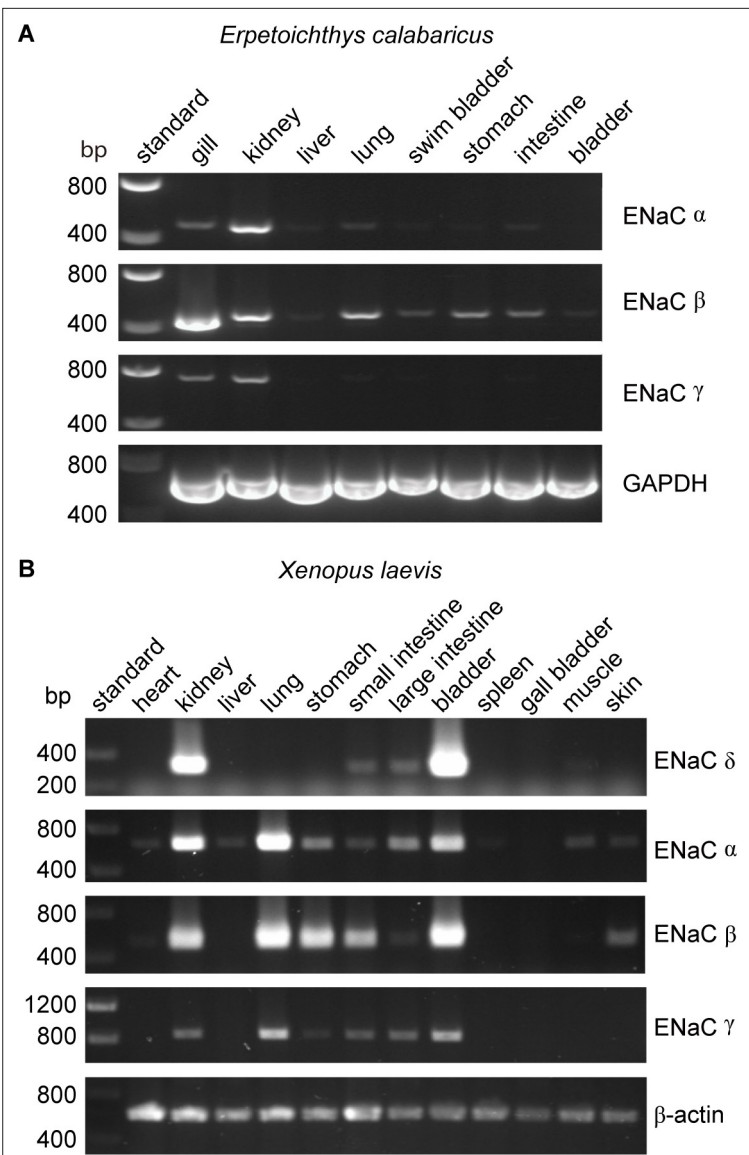

**Figure 4.** Tissue distribution of (A) *Erpetoichthys calabaricus* (ropefish) and (B) *Xenopus laevis* epithelial Na⁺ channel (ENaC) subunit transcripts by reverse transcription-PCR (RT-PCR). cDNA libraries were generated from tissue homogenates. PCR reactions were performed using primers indicated in *Supplementary file 3*.

The online version of this article includes the following source data for figure 4:

**Source data 1.** Uncropped gels are shown with bands at the target size for a given primer pair indicated.

investigation of α and δ subunit expression in *X. laevis* tissues (*Wichmann et al., 2018*), and suggest that cleavage regulates ENaC function in both the lungs and kidneys of frogs, similar to mammals.

## α and γ subunit GRIP domain cleavage sites evolved independently

The absence of cleavage sites in species rooted closest to the early gene duplication events suggests that each cleavage site did not result from a common ancestor, but rather appeared independently in the α and γ subunits. To test this idea, we performed ancestral reconstruction of nodes 1–4 in our phylogenetic tree (*Figure 2*) using BayesTraits and MCMC methods. In the unconstrained evolutionary model, each of these nodes was free to adopt any status for the site of interest. In the divergent evolutionary model, nodes 1–4 were constrained to contain the site of interest. In the convergent evolutionary model, node 1 was constrained to exclude the site of interest. Models were compared by converting marginal likelihoods resulting from MCMC runs of each model to log Bayes Factors, where

**Table 2.** PY motifs (L/P-P-X-Y) in the C-terminal tails of epithelial Na⁺ channel (ENaC) subunits from various species.

Ø, no sequence was available. (⋯), sequence for the C-terminal region was missing in the available sequence. Species abbreviations are as in *Figure 2*.

| Animal | Terrestrial | Lungs | α | δ | β | γ |
|---|---|---|---|---|---|---|
| S.Lamprey | | | PPPSF | Ø | PPPHY | PPPQY |
| J.Lamprey | | | PPDY | Ø | PPPHY | PPPQY |
| E.Shark | | | (⋯) | Ø | PPPRY | PPPNY |
| Ropefish | | X | PPPAY | Ø | PPPHY | PPPNY |
| Coelacanth | | | PPAY | (⋯) | PPPNY | PPPTY |
| W.Lungfish | | X | PPPAY | Ø | PPPHY | PPPQY |
| A.Lungfish | | X | PPPAY | Ø | PPPKY | PPPQY |
| Frog | X | X | PPPAY | — | PPPNY | PPPKY |
| Salamander | X | X | PPPAY | Ø | Ø | Ø |
| Turtle | X | X | LPSY | — | PPPNY | PPPNY |
| Chicken | X | X | LPSY | — | PPPNY | PPPNY |
| Cow | X | X | PPPAY | — | PPPNY | PPPRY |
| Human | X | X | PPPAY | — | PPPNY | PPPKY |

values of >2, 5–10, and >10 support positive, strong, and very strong preferences for the better fitting model, respectively. Within the framework of sites 1 and 2 coevolving with each other, comparison of the unconstrained evolutionary model with the convergent evolutionary model resulted in log Bayes Factors of 0.38 for site 1 and 0.34 for site 2, supporting no preference between models. Comparison of the divergent evolutionary model to the convergent model for each of the sites resulted in log Bayes Factors of 6.5 in both cases, supporting a strong preference for the convergent evolutionary model for each of the cleavage sites. This suggests that, despite analogous molecular mechanisms in the α and γ subunits, the appearance of cleavage sites in the α and γ subunits were independent events. Neither the proximal nor the distal cleavage site arose from a common pre-α/γ ENaC subunit.

### ENaC PY motifs evolved through divergent evolution

Our data suggest that cleavage of the α and γ subunits evolved contemporaneously with aldosterone synthase. Indeed, aldosterone enhances activating cleavage of ENaC subunits (*Frindt and Palmer, 2015*; *Terker et al., 2016*). Due to the apparent connection to aldosterone, we examined another ENaC target of aldosterone-dependent regulation, the PY motifs in the C-termini. The PY motifs facilitate enhanced channel turnover via Nedd4-2, which can be inhibited by aldosterone and other hormones through kinase signaling (*Rotin and Staub, 2012*). Mutation or deletion of the PY motifs increases ENaC function, leading to Liddle syndrome which is characterized by hypertension, hypokalemia, and low aldosterone levels (*Cui et al., 2017*). We found PY motifs in all α, β, and γ subunits where sequences of the C-termini were available (*Table 2*, *Figure 1—source data 1*), suggesting a divergent evolutionary model that did not depend on terrestrial status. Interestingly, δ subunits apparently lost their PY motifs after their divergence from the α subunit in the coelacanth ancestor. We tested whether appearance of the PY motif depended on the terrestrial migration or the development of lungs using nested likelihood models, as above. A likelihood ratio test between the nested models showed no preference for the dependent model for either terrestrial status (p = 0.54) or lungs (p = 0.28), suggesting their evolution was affected by neither. To test whether the PY motifs evolved from a common ancestor, we tested models of PY motif evolution using ancestral reconstruction at nodes 1–4 in *Figure 2* and MCMC methods. The unrestricted model suggested probabilities of a PY motif present at 98.1%, 99.3%, 91.5%, and 75.9% for nodes 1, 2, 3, and 4, respectively. We then tested a series of restricted models to test competing evolutionary models. In the divergent evolutionary model, nodes 1–4 were restricted to contain PY motifs. Comparison of the unrestricted and

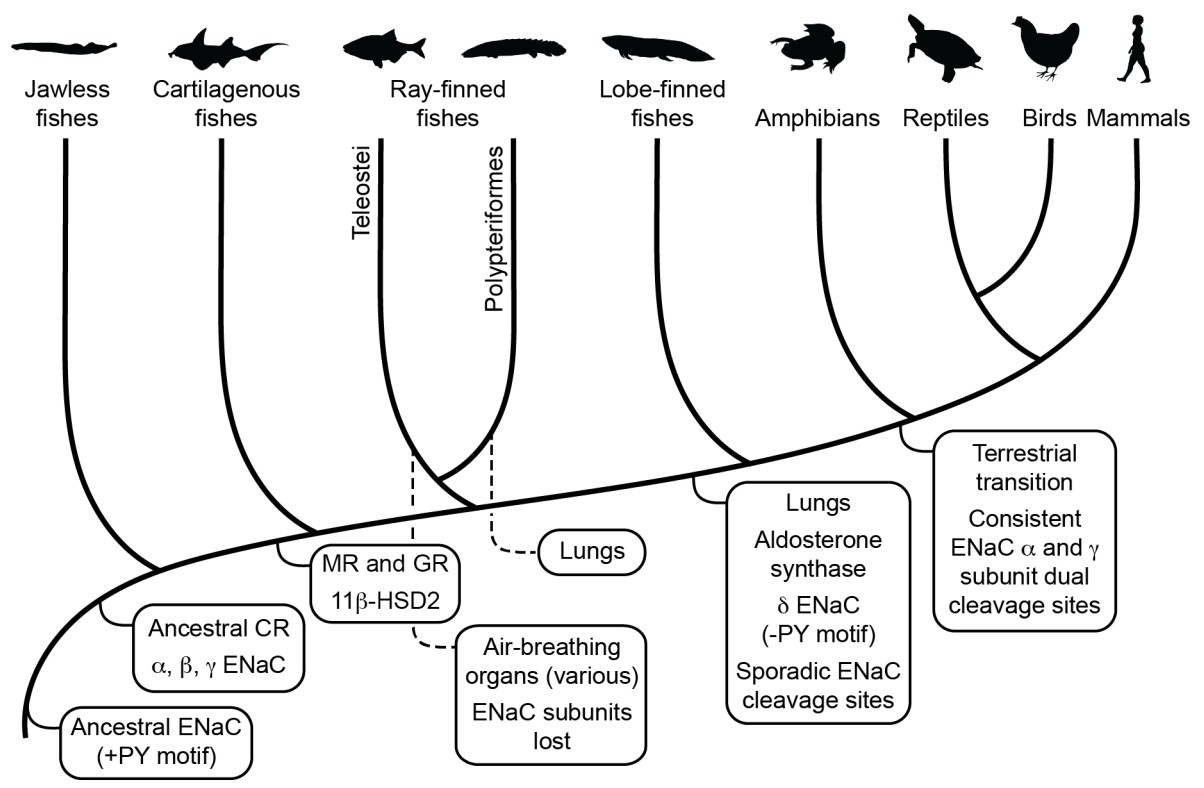

**Figure 5.** Schematic view of evolution of aldosterone signaling, air-breathing organs, epithelial Na+ channel (ENaC), and ENaC regulatory motifs. ENaC subunits were not found in nonvertebrate chordates or in teleosts. The ancestral ENaC subunit likely had a PY motif and was a substrate for Nedd4-2-dependent regulation. Like mammalian ENaC α subunits, the ancient ENaC subunit may have formed functional homotrimers, or may have formed channels with other ENaC paralogs. ENaC α, β, and γ subunits appeared before the emergence of jawless fishes, whereas ENaC δ subunits first appeared in an ancestor of the lobe-finned coelacanth. Proteins required for aldosterone signaling (mineralocorticoid receptor [MR], 11β-hydroxysteroid dehydrogenase [11β-HSD2], and aldosterone synthase) evolved before the emergence of tetrapods. Lungs appeared in lobe-finned fishes on the lineage to tetrapods. Air-breathing organs (e.g., respiratory gas bladders and labyrinth organs) evolved independently in ray-finned fishes, including morphologically distinct lungs in Polypteriformes. Individual GRIP (gating release of inhibition by proteolysis) domain ENaC cleavage sites first appeared sporadically in marine species. Dual cleavage sites appeared consistently in the ENaC α and γ subunits in terrestrial vertebrates. CR, corticoid receptor; GR, glucocorticoid receptor. Animal silhouettes courtesy of PhyloPic (http://www.phylopic.org).

divergent evolutionary models resulted in a log Bayes Factor of 0.19, suggesting these models were nearly equivalent. In convergent evolutionary models, one of the four early nodes was restricted to not contain a PY motif, resulting in four models. Comparison of the divergent model to each of the convergent models resulted in log Bayes Factors ranging from 4.7 to 8.9, suggesting a strong preference for the divergent model over each of the nondivergent models. These data suggest that the PY motif in each subunit, in contrast to the polybasic tracts, arose from a common progenitor.

## Discussion

Our results suggest that the two ENaC GRIP domain cleavage sites in each of the α and γ subunits coevolved with each other, but appeared independently in each subunit. Our results also suggest that double cleavage coevolved with both the terrestrial migration of vertebrates and the development of lungs (*Figure 5*). Coevolution of site 1 with site 2 makes sense given that functional consequences of cleavage in ENaCs only result from double cleavage of a single subunit followed by liberation of the intervening inhibitory tract (*Bruns et al., 2007*; *Carattino et al., 2006*). We found that the polybasic tracts appeared sporadically in the GRIP domain of ENaC subunits from marine species, but appeared and remained consistently in terrestrial vertebrates. Terrestrial life required adaptive changes to overcome the stresses of electrolyte regulation and gas exchange. Coevolution of the terrestrial migration

with tandem GRIP domain polybasic tracts, and consequently higher ENaC activity, suggests that selection pressure could have arisen from either stressor.

Air-breathing organs evolved independently several times as an adaptation to chronic or periodic environmental hypoxia (*Graham and Wegner, 2010*). Lungs appeared in lobe-finned fishes after their divergence from other bony fishes, and developmental and morphological evidence support homology between the lungs of lungfishes and tetrapods. Extant deep-water coelacanths possess vestigial lungs that correspond to the likely functional lungs of the Cretaceous shallow-water coelacanth *Axelrodichthys* (*Cupello et al., 2015*; *Brito et al., 2010*). Air-breathing organs evolved several times in ray-finned fishes, giving *Polypteridae* (ropefish) morphologically distinct lungs, and other ray-finned fishes air-breathing capabilities through modified gas bladders and labyrinth organs (*Figure 5*). Notably, we identified ENaC subunits in ropefish and not in any other ray-finned fish, suggesting ENaC subunits were lost soon after the divergence of Polypteriformes from ray-finned fishes in the Devonian period ~360 million years ago (*Hughes et al., 2018*). Their loss reflects a lack of selective pressure to maintain the ENaC subunits. Evidence supports electrogenic Na$^+$ transport through an ENaC-like channel in the gills of ray-finned fishes (*Evans et al., 2005*). Related proteins like ASIC4, proposed to play this role (*Dymowska et al., 2014*), or the uncharacterized ENaC-like paralogs may have left ENaC subunits dispensable. Airway surface liquids are critical to the function of mammalian lungs and are regulated by ENaC and other ion channels (*Haq et al., 2016*). ENaC α subunit knockout mice die of asphyxiation shortly after birth due to an inability to clear fluid from the lung (*Hummler et al., 1996*), and ENaC polymorphisms have been associated with lung dysfunction (*Rauh et al., 2010*; *Schaedel et al., 1999*). ENaC regulation by cleavage is relevant in primary airway epithelial cell cultures (*Reihill et al., 2016*; *Myerburg et al., 2010*), where ENaC is regulated by glucocorticoids rather than by aldosterone (*Stokes and Sigmund, 1998*). However, transcripts of ENaC subunits with GRIP domain polybasic tracts were not detected in the lungs of lungfishes (*Uchiyama et al., 2015*; *Uchiyama et al., 2012*) and were faint in the lungs of the ropefish (*Figure 4*). As lungs coevolved with the terrestrial migration of vertebrates, concurrent stresses may have provided selection pressure for ENaC activation by cleavage.

Emersion from an aquatic environment poses a risk of desiccation and a challenge to terrestrial life (*Wright and Turko, 2016*). Aldosterone signaling through MR regulates electrolyte balance and total body volume in tetrapods. Key molecules required for aldosterone signaling appeared at various points during the evolution of vertebrates (*Figure 5*; *Rossier et al., 2015*). Interestingly, aldosterone synthase is absent in cartilaginous and ray-finned fishes, but appears in lobe-finned fishes on the lineage to terrestrial vertebrates (*Joss et al., 1994*). The appearance of aldosterone synthase coincided with the appearance of the ENaC δ subunit, and preceded the consistent appearance of tandem polybasic tracts in the ENaC α and γ subunits. In ray-finned fishes, osmoregulation and euryhaline adaptation primarily occur through glucocorticoid receptor signaling (*Takahashi and Sakamoto, 2013*). Teleost MR signaling is implicated in central nervous system functions that include sodium appetite, but plays a secondary role in ion transporter regulation in osmoregulatory organs, in contrast to MR signaling in terrestrial vertebrates. Notably, aldosterone promotes cleavage of the ENaC α and γ subunits in mammals through MR signaling (*Frindt and Palmer, 2015*; *Terker et al., 2016*). Analysis of transcripts from ropefish and lobe-finned lungfishes show expression of ENaC α, β, and γ subunit transcripts in the gills and kidneys (*Figure 4*), which are important sites of ion regulation in fishes (*Uchiyama et al., 2015*; *Uchiyama et al., 2012*). Furthermore, ENaC α subunit mRNA levels were positively correlated with plasma aldosterone in the West African lungfish *Protopterus annectins*, suggesting an important role for aldosterone and ENaC in electrolyte homeostasis in lobe-finned fishes (*Uchiyama et al., 2015*).

Why do ENaC subunits have polybasic tracts? Integral membrane proteins including ENaC are further processed and sorted in the Golgi apparatus for transport to their destinations. Furin and other proprotein convertases are present in various parts of the Golgi complex and cleave after polybasic motifs where Arg is preferred in the last position (*Seidah and Prat, 2012*). Selective double cleavage strongly increases ENaC currents, providing the functional change required for fitness-based selection. ENaC processing through the Golgi apparatus, where furin or other proprotein convertases are present, may have led to the emergence of polybasic tracts. Furin homologs are present in *Drosophila*, consistent with furin emerging well before ENaC subunits, and before the divergence of deuterostomes and protostomes (*Roebroek et al., 1991*; *Bertrand et al., 2006*). Prostasin, also

known as PRSS8, cleaves the distal site of the γ subunit in mammals. Prostasin may have emerged in amphibians (*Bhagwandin et al., 2003*) as the most prostasin-related proteins in nontetrapods are more closely related to other mammalian proteases (PRSS22 and PRSS27). Notably, the last residue of the γ subunit distal site switched from Arg in amphibians to Lys in other tetrapods (*Table 1*, γ – site 2). Furin has a requirement Arg in the last position (*Henrich et al., 2003*) while prostasin requires either Arg or Lys in the last position (*Shipway et al., 2004*). This Arg to Lys switch prevents furin cleavage in the *trans*-Golgi network, and allows aldosterone-inducible prostasin to cleave and activate the channel at the cell surface.

The evolution of ENaC GRIP domain cleavage sites contrasts with the divergent evolution of PY motifs in the C-termini, which appear in all extant α, β, and γ subunits, but was lost in δ subunits that first appeared in a coelacanth ancestor. The presence of both ENaC subunit PY motifs and Nedd4-like proteins in ancestors of jawless fishes (*Harvey and Kumar, 1999*) suggests that aldosterone signaling co-opted Nedd4-2-dependent regulation. Differences in physiologic roles may underlie loss of the PY motif in ENaC δ subunits. ENaC δ subunit tissue distribution aligns poorly with aldosterone sensitivity, and is different than for the other ENaC subunits (*Giraldez et al., 2012*). Despite the lack of PY motifs, δ subunits can be ubiquitylated by Nedd4-2, presumably through binding β or γ subunit PY motifs (*Kevin et al., 2013*). This is possible where subunit expression overlaps (e.g., lung and esophagus), but there are neuronal and reproductive tissues where only the δ subunit has been detected.

In summary, we found that two forms of ENaC regulation modulated by aldosterone signaling emerged through distinct evolutionary patterns. Channel activation by double cleavage of the α and γ subunits coevolved with the terrestrial migration of vertebrates, closely paralleling that of aldosterone synthase and likely as a mechanism to enhance Na+ conservation and fluid retention. Regulation of cell surface stability through PY-motif interactions with Nedd4-2 was likely present in Cambrian period ancestral ENaC subunits and preceded the development of aldosterone signaling.

## Materials and methods
### Sequence retrieval, alignment, and phylogenetic tree calculation
Human ENaC subunit protein sequences were used as query sequences in BLAST searches of databases (see *Supplementary file 1*). Organism restrictions and organism specific databases were used to identify proteins in each of the major vertebrate classes. Alignment using MUSCLE 3.8.31 (*Edgar, 2010*), curation using Gblocks 0.91b (*Talavera and Castresana, 2007*), and phylogenetic tree calculation using PhyML (*Guindon et al., 2010*) were performed using Phylogeny.fr (*Dereeper et al., 2008*). The final tree (*Supplementary file 4*) was calculated using the *smaller final blocks* option in Gblocks and the *bootstrapping procedure* option (100 bootstraps) in PhyML. Residue frequency in the aligned sequences was visualized using WebLogo (*Crooks et al., 2004*). Trees were visualized using FigTree (http://tree.bio.ed.ac.uk/software/figtree/).

### Motif analysis
Furin cleaves human α ENaC after RSRR[178] and RRAR[204], and human γ ENaC at RKRR[138] (*Hughey et al., 2004a*). Furin requires Arg at P1 and has a preferred P4–P3–P2–P1↓ = RX-R/K-R↓ substrate sequence, although deviations at P2 and P4 have been observed, for example Ala-203 at P2 in the ENaC α subunit (*Duckert et al., 2004*; *Tian et al., 2011*). Prostasin cleaves the human ENaC γ subunit after RKRK[181] (*Bruns et al., 2007*). Prostasin cleavage requires Arg or Lys at P1, and prefers basic residues at P2, P3, and P4 (*Shipway et al., 2004*). We inspected the region in the alignment that contains the human ENaC cleavage sites (*Figure 1—source data 1*) for ideal furin sites or alternatively, for polybasic tracts ending in Arg that aligned within five residues of either of the α subunit furin sites for α and δ subunits, the γ subunit furin site for β and γ subunits, or for any of the furin sites for ASIC subunits. We also inspected the alignment for polybasic tracts ending in Arg or Lys that aligned within five residues of the human γ subunit prostasin site for β and γ subunits. For the PY motif (L/P-P-X-Y) required for Nedd4-2-dependent regulation (*Persaud et al., 2009*), we inspected the C-terminal region of the alignment. All sequences found aligned with human ENaC subunit PY motifs in the C-termini.

## Testing phylogenetic models of site gain and Loss

BayesTraits V3 (*Meade, 2022*; *Pagel et al., 2004*) was used to compare evolutionary models of two binary traits using maximum likelihood methods, as described above. Traits were assigned as indicated in *Tables 1 and 2*: absent, present, or ambiguous (in cases where the relevant region was missing). ASICs in our dataset were all assigned 'marine', and only coelacanth ASIC2 was assigned as having a polybasic tract at site 2 (*Figure 1—source data 1*). BayesTraits run parameters are provided in *Supplementary file 2* and results are provided in *Source data 1*. Nested models were compared using a likelihood ratio test, with degrees of freedom equal to the difference in the number of parameters for each model. The likelihood ratio statistic is calculated as: likelihood ratio = 2 × {log-likelihood(dependent model) − log-likelihood(independent model)}, and was converted to a p value using the *chisq.dist.rt* function of Excel. BayesTraits was also used to determine whether traits likely appeared independently (convergent model), or derived from a common ancestor (divergent model). Program parameters are provided in *Supplementary file 2* and results are provided in *Source data 1*. Values of a given trait at all key ancestral nodes were determined while trait values at specific nodes were fixed to reflect each of the hypothetical evolutionary models. Maximum likelihood runs of each model were used to determine likely average values for model parameters. Uniform priors were selected for parameters with large expected values, and exponentially distributed priors with a mean of 0.001 were selected for parameters with small expected values. MCMC runs for each model were performed for the default number of iterations (1,010,000), and generated model parameters with means similar to those from maximum likelihood runs, consistent with convergence. The stepping stone sampler (100 stones with 10,000 iterations) was used to calculate log marginal likelihood values from MCMC runs. Log marginal likelihood values were converted to log Bayes Factors {log Bayes Factors = 2 × (log marginal likelihood model 1 − log marginal likelihood model 2)} for model comparisons. Values of log Bayes Factors were used as evidence for model preference, where <2 was weak evidence, >2 was positive evidence, 5–10 was strong evidence, and >10 was very strong evidence.

## Tissue distribution of ENaC subunit transcripts

All animals were handled according to approved institutional animal care and use committee protocols (#20037084 and #21018704) of the University of Pittsburgh. Total RNA was extracted from various tissues isolated from *E. calabaricus* and *X. laevis* (RRID:XEP_Xla100) using TRIzol Reagent (Invitrogen) and denatured by heating at 70°C for 10 min. Using 2 µg of RNA isolated from each tissue, cDNA libraries were synthesized using RevertAid First Strand cDNA Synthesis Kit (Thermo Fisher). Specific primers for the frog ENaC α, β, and γ subunits, each of the ropefish ENaC subunits, and glyceraldehyde-3-phosphate dehydrogenase (GAPDH) as an internal standard (*Supplementary file 3*) were designed using the NCBI Primer-Blast tool (https://www.ncbi.nlm.nih.gov/tools/primer-blast/). Primers for frog δ ENaC subunits and β-actin were previously described (*Wichmann et al., 2018*). All primers were custom synthesized (Integrated DNA Technologies). PCR was run for 30 cycles using GoTaq G2 Green master mix (Promega) and specific primers, with the annealing temperature set at 60°C. PCR products were visualized after agarose gel electrophoresis using GelRed stain (Sigma) and a GelDoc imaging system (BioRad).

Plasmids and site-directed mutagenesis cDNAs encoding mouse ENaC subunits were previously described (*Shimkets et al., 1997*; *Passero et al., 2010*; *Hughey et al., 2003*). Australian lungfish γ subunit with a C-terminal epitope tag (Aγ) was synthesized by Twist Bioscience (San Francisco, CA) and cloned into pcDNA 3.1 hygro (+). Site-directed mutagenesis of Aγ was performed using the QuikChange II XL Site-directed Mutagenesis Kit (Agilent, Santa Clara, CA). cRNAs were transcribed using mMessage mMachine Transcription Kits (Invitrogen), and purified using the RNeasy MiniElute Cleanup Kit (Qiagen).

## ENaC expression in *Xenopus* oocytes

Oocytes were harvested from *X. laevis* following a protocol approved by the University of Pittsburgh Institutional Animal Care and Use Committee, as previously described (*Wang et al., 2019*). Stage V–VI oocytes were injected with 4 ng of cRNA per ENaC subunit, as indicated. Injected oocytes were maintained in modified Barth's saline (MBS: 88 mM NaCl, 1 mM KCl, 2.4 mM $NaHCO_3$, 0.3 mm $Ca(NO_3)_2$, 0.41 mM $CaCl_2$, 0.82 mM $MgSO_4$, and 15 mM HEPES, pH 7.4) supplemented with 10 µg/ml sodium penicillin, 10 µg/ml streptomycin sulfate, and 100 µg/ml gentamycin sulfate at 18°C for 24 h.

## Current measurement by two-electrode voltage clamp

Oocytes were mounted in a continuously perfused 20 µl recording chamber and impaled by two 0.1–1 MΩ recording pipettes filled with 3 M KCl. Voltage was clamped at −100 mV and currents were continuously recorded using an Axoclamp 900A voltage clamp amplifier (Molecular Devices, Sunnyvale, CA) and pClamp 10.5 software (Molecular Devices). Baseline currents were measured in Na-110 buffer (110 mM NaCl, 2 mM KCl, 2 mM $CaCl_2$, and 10 mM HEPES, pH 7.4). Amiloride (Na-110 supplemented with 100 µM amiloride) was added at the end of each experiment to block ENaC and determine the ENaC-mediated portion of the current. Statistical analysis of current data was performed using Prism 9 (GraphPad Software).

## Surface biotinylation and western blotting

One day after injection, oocytes were transferred to a 12-well dish and incubated in MBS on ice for 30 min. After washing twice, surface proteins were labeled with biotin by adding 1 mg/ml membrane impermeant EZ-Link Sulfo-NHS-SS-Biotin (Thermo Fisher) in 137 mM NaCl and 15 mM sodium borate, pH 9.0 for 30 min. Excess reagent was quenched with MBS supplemented with 192 mM glycine. After two washes with MBS, approximately 40 oocytes were lysed in 500 µl detergent solution (100 mm NaCl, 40 mm KCl, 1 mm EDTA, 10% glycerol, 1% NP-40, 0.4% deoxycholate, 20 mm HEPES, pH 7.4, supplemented with protease inhibitor mixture III [Calbiochem]). After reserving 2% of the whole cell lysate, the remainder was incubated with NeutrAvidin agarose (Thermo Fisher) overnight at 4°C on a rocker to isolate biotinylated proteins. Biotinylated proteins were eluted by boiling in 2× Laemmli buffer (BioRad). Biotinylated proteins representing the total and surface pools were separated by SDS–PAGE (4%–15% Tris/glycine, BioRad) and blotted for the γ subunit using anti-HA-Peroxidase antibodies (1:2000, RRID:AB_439705) or GAPDH using anti-GAPDH-Peroxidase antibodies (1:10,000, RRID:AB_1078992). Band densities were measured using a ChemiDoc Imaging system (BioRad). Statistical analyses were performed using Prism 9.

## Acknowledgements

This work was supported by NIDDK, National Institutes of Health, Grant R01 DK125439 (to O.B.K.). The Pittsburgh Center for Kidney Research was supported by P30DK079307 from NIDDK, and the Pittsburgh Liver Research Center was supported by P30DK120531 from NIDDK.

## Additional information

### Funding

| Funder | Grant reference number | Author |
| --- | --- | --- |
| National Institute of Diabetes and Digestive and Kidney Diseases | Grant R01 DK125439 | Ossama B Kashlan |

The funders had no role in study design, data collection, and interpretation, or the decision to submit the work for publication.

### Author contributions

Xue-Ping Wang, Formal analysis, Investigation, Methodology, Writing - original draft, Writing - review and editing; Deidra M Balchak, Investigation; Clayton Gentilcore, Data curation, Formal analysis, Writing - review and editing; Nathan L Clark, Conceptualization, Formal analysis, Investigation, Methodology, Supervision, Writing - original draft, Writing - review and editing; Ossama B Kashlan, Conceptualization, Data curation, Formal analysis, Funding acquisition, Investigation, Methodology, Project administration, Resources, Supervision, Validation, Visualization, Writing - original draft, Writing - review and editing

### Author ORCIDs

Ossama B Kashlan http://orcid.org/0000-0003-0537-6720

### Ethics

This study was performed in strict accordance with the recommendations in the Guide for the Care and Use of Laboratory Animals of the National Institutes of Health. All of the animals were handled according to approved institutional animal care and use committee (IACUC) protocols (#20037084 and #21018704) of the University of Pittsburgh. All surgery was performed following tricaine methane sulfonate anesthesia and sacrifice, and every effort was made to minimize suffering.

### Decision letter and Author response

Decision letter https://doi.org/10.7554/eLife.75796.sa1
Author response https://doi.org/10.7554/eLife.75796.sa2

## Additional files

### Supplementary files

• Supplementary file 1. Protein sequences were found using BLAST tools at NCBI (https://www.ncbi.nlm.nih.gov/) and UniProt (https://www.uniprot.org/). #Annotated as epithelial Na+ channel (ENaCα) subunit in NCBI, but named ASIC1 here on the basis of calculated phylogenetic tree in *Figure 2*. *Protein sequences were originally found using the BLAST tool at A*STAR (http://jlampreygenome.imcb.a-star.edu.sg/ and http://esharkgenome.imcb.a-star.edu.sg/), but were no longer available at the time of publication. Coding sequences are available at the accession numbers shown at NCBI.

• Source data 1. Results of phylogenetic models of trait gain or loss are provided in a Microsoft Excel file.

• Supplementary file 2. BayesTraits run parameters. [1]ML, maximimum likelihood run. All ML runs included the commands: MLTries 10,000; ScaleTrees 1000. [2]Markov chain Monte Carlo run. All MCMC runs included the commands: ScaleTrees 1000; Stones 100, 10,000. [3]Restricts reverse rate for trait 1–0. [4]Restricts trait one forward rates to be independent of trait 2, and trait 1 reverse rates to 0. [5]Restricts the equivalent sites 1 and 2 rates to be equal. [6]Restricts the equivalent sites 1 and 2 rates to be equal, but dependent on the status of the other site. [7]The AddTag and Fossil commands select and restrict a node given by the most recent common ancestor of the proteins specified. Nodes 1, 2, 3, and 4 in *Figure 2* are specified in commands as 'ENaC', 'bg', 'alpha', and 'ad', respectively.

• Supplementary file 3. Primers for RT-PCR and expected product sizes.

• Supplementary file 4. Tree file.

• Transparent reporting form

### Data availability

Source data for figures 1 and 2 are provided in Supplementary file 1. Source data for trait evolution analysis, Figures 3 and Figure 4 are provided as image and Microsoft Excel files in Source data 1. Images and source data of electrophysiology traces in Figure 3 are provided at the Zenodo data repository (https://doi.org/10.5281/zenodo.5790375).

The following dataset was generated:

| Author(s) | Year | Dataset title | Dataset URL | Database and Identifier |
|---|---|---|---|---|
| Kashlan OB | 2021 | Figure 3-source data 2 | https://zenodo.org/record/5790375 | Zenodo, 10.5281/zenodo.5790375 |

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
