## [Editor Report]

The authors show that enzymatic regulation of a sodium-permeable channel that aids in gas exchange and fluid homeostasis likely evolved at the water to land transition in vertebrates. This work is impactful as it details a critical period in the evolution of terrestrial vertebrates.

---

## [Decision Letter]

**Decision letter after peer review:**

[Editors’ note: the authors submitted for reconsideration following the decision after peer review. What follows is the decision letter after the first round of review.]

Thank you for submitting your work entitled "Activation by Cleavage of the ENaC α and γ Subunits Independently Coevolved with the Vertebrate Terrestrial Migration" for consideration by *eLife*. Your article has been reviewed by 3 peer reviewers, including Lauren A O'Connell as the Reviewing Editor and Reviewer #1, and the evaluation has been overseen by a Senior Editor. The following individual involved in review of your submission has agreed to reveal their identity: Harold H Zakon (Reviewer #2).

Our decision has been reached after consultation between the reviewers. Based on these discussions and the individual reviews below, we regret to inform you that your work will not be considered further for publication in *eLife*.

The reviewers agreed that the findings presented in this manuscript present a great evolutionary story about ENaC channels in the water to land transition of vertebrates. While the manuscript presents many novel findings, these claims were not supported by functional evidence. We encourage the authors to either add functional evidence from the literature or from follow-up laboratory experiments. We felt the experiments required for functional validation were longer than an appropriate revision period during COVID-19.

*Reviewer #1:*

As someone who is not familiar with ENaC channels, I found the manuscript esoteric and difficult to read. I encourage the authors to simplify their writing to appeal to the broad readership of *eLife*.

I am confused as to what the "uncharacterized ENaC-like proteins" in fish were. How was the decision made to include these are not and what are these transcripts?

The authors mention a ENaC knockout in mice, but has this also been done in *Xenopus*? Since they transition from water to land within their life cycle, it would seem a useful test of this idea, although I do not think this is required for publication.

*Reviewer #2:*

This paper highlights a series of molecular events in vertebrate evolution that occurred during the period preceding the transition to land of tetrapods. Specifically, it focuses on the evolution of the ENac channels that regulate sodium levels in the kidney and shows that α and γ ENac subunits gained two proteolytic cleavage sites concurrently with the evolution of lungs and transition to land and, importantly, with the evolution of an enzyme that makes aldosterone, a hormone that regulates the numbers of ENac channels. Additionally, in tracing the phylogeny of ENac channels in chordates, this paper shows that ENac channel genes were lost in Osteichthyes following the divergence of polypteriformes and the other Osteichthyes, most importantly, the teleosts. Although the authors do not discuss this, it would be interesting to know if the loss of these genes is related to the evolutionary origin of teleosts in fresh water.

This makes a tidy story. This is a well-written and clearly illustrated paper. I particularly like the summary figure (Figure 4).

The consensus sequence of the polybasic cleavage sites varies across tetrapods (Table 1). One concern I have is that it is assumed that all of these polybasic sequences support cleavage. What is known about the actual cleavage of these in vertebrates other than mammals? Have the ENac channels from frogs, say, been tested with mammalian (if not frog) enzymes to know that these polybasic sites are cleavable? It is concerning to read that "The human d subunit sequence also exhibits a polybasic tract in this region, but experimental evidence shows that it is not cleaved." I would like to see some evidence from the literature or data from the authors demonstrating that the putative cleavage in non-mammalian proteins really occurs at these sites. It is possible they evolved for another reason initially and later became cleavage sites.

*Reviewer #3:*

Wang et al. investigate the development of a protease-dependent regulatory mechanism of ENaC function. They specifically look into the emergence of polybasic tracts in the α and γ subunits. Furthermore, they compare transcript levels of ENaC subunits between ropefish and African clawed frog where the former independently developed lungs from tetrapods and the latter evolved after transitioning to terrestrial habitat. As it currently stands, this manuscript is focused on one topic, essentially the emergence of polybasic tract in ENaC subunits and is not well suited for a broader readership that *eLife* supports. There are a few comments that authors can expand on and address to create a manuscript that is more wide-reaching in terms of reader interests.

1) As the authors surely know, there are other protease recognition sites that do not necessarily contain polybasic tracts. Why do ENaC subunits have polybasic tracts? Is it due to the protease furin? When did furin emerge?

2) Related to the first comment, are there other protease sites in any of the ENaC subunits that are not polybasic? When did they emerge?

3) The authors provide RT-PCR data of ENaC subunits in different tissues derived from ropefish and African clawed frog. Can the authors explain the result in Figure 3 as to why there is no considerable signal of α and γ subunits, only faint bands of δ and β from frog skin? There are numerous functional data – seminal studies of ENaC, in fact – collected from frog skin that suggest a large role of ENaC in regulating Na^+^ flux. Does the subunit composition on frog skin ENaC lack α and γ subunits?

---

## [Author Response]

[Editors’ note: the authors resubmitted a revised version of the paper for consideration. What follows is the authors’ response to the first round of review.]

Reviewer #1:As someone who is not familiar with ENaC channels, I found the manuscript esoteric and difficult to read. I encourage the authors to simplify their writing to appeal to the broad readership of eLife.I am confused as to what the "uncharacterized ENaC-like proteins" in fish were. How was the decision made to include these are not and what are these transcripts?

We believe these sequences are important to include to establish that ENaC subunits first appear in jawless fishes, and that ENaC subunits were lost in ray finned fishes after the divergence of Polypteriformes. Searches restricted to ray finned fishes only returned ENaC subunits in early branching Polypteriformes. Other top hits were either ASIC subunits or proteins annotated as "ENaC-like". Searches restricted to non-vertebrate chordates also returned proteins annotated as ENaC-like proteins. Our phylogenetic analysis revealed that the ENaC-like proteins from lancelets and ray finned fishes belong to a clade distinct from ENaC and ASIC subunits. To address a question raised by Reviewer 2, we revised our manuscript to briefly discuss the ENaC-like proteins (page 11, line 31).

The authors mention a ENaC knockout in mice, but has this also been done in *Xenopus*? Since they transition from water to land within their life cycle, it would seem a useful test of this idea, although I do not think this is required for publication.

Cleavage and activation of *Xenopus* ENaC through the α and γ subunits was previously reported. We now cite this work in the results subsection describing the emergence of polybasic tracts (page 6, line 13). We have also included new experiments examining cleavage of ENaC γ subunit from the Australian lungfish, which we predicted to be cleaved. These data are consistent with activating cleavage at the predicted polybasic sites in the GRIP domain (Figure 3 and page 6, line 29). ENaC subunits have never been knocked out in *Xenopus* to our knowledge, and while this is an interesting experiment, it is beyond the scope of this study.

Reviewer #2:This paper highlights a series of molecular events in vertebrate evolution that occurred during the period preceding the transition to land of tetrapods. Specifically, it focuses on the evolution of the ENac channels that regulate sodium levels in the kidney and shows that α and γ ENac subunits gained two proteolytic cleavage sites concurrently with the evolution of lungs and transition to land and, importantly, with the evolution of an enzyme that makes aldosterone, a hormone that regulates the numbers of ENac channels. Additionally, in tracing the phylogeny of ENac channels in chordates, this paper shows that ENac channel genes were lost in Osteichthyes following the divergence of polypteriformes and the other Osteichthyes, most importantly, the teleosts. Although the authors do not discuss this, it would be interesting to know if the loss of these genes is related to the evolutionary origin of teleosts in fresh water.

We agree this is an interesting question, but we just don’t know the answer to this. One explanation could be that another channel made ray finned ENaC subunits superfluous. We revised our manuscript to discuss this possibility (page 11, line 31).

“Evidence supports electrogenic Na^+^ transport through an ENaC-like channel in the gills of ray finned fishes (15618479). Related proteins like ASIC4 (24898589), proposed to play this role, or the uncharacterized ENaC-like proteins may have left ENaC subunits dispensable.”

This makes a tidy story. This is a well-written and clearly illustrated paper. I particularly like the summary figure (Figure 4).The consensus sequence of the polybasic cleavage sites varies across tetrapods (Table 1). One concern I have is that it is assumed that all of these polybasic sequences support cleavage. What is known about the actual cleavage of these in vertebrates other than mammals? Have the ENac channels from frogs, say, been tested with mammalian (if not frog) enzymes to know that these polybasic sites are cleavable? It is concerning to read that "The human d subunit sequence also exhibits a polybasic tract in this region, but experimental evidence shows that it is not cleaved." I would like to see some evidence from the literature or data from the authors demonstrating that the putative cleavage in non-mammalian proteins really occurs at these sites. It is possible they evolved for another reason initially and later became cleavage sites.

Cleavage of ENaC α and γ subunits from a frog (*Xenopus laevis*) has been confirmed, and proved to be functional (29576549). We revised our manuscript to highlight this result in the context of GRIP domain cleavage (page 6, line 13). We also performed new experiments to examine the ENaC g subunit from Australian lungfish expressed in *Xenopus oocytes*. Our data show cleavage of Australian lungfish γ ENaC at the predicted sites, and associated channel activation. These data are presented in Figure 3, and a new results subsection titled, Australian lungfish ENaC g subunit is cleaved at GRIP domain polybasic tracts (page 6, line 29)*.*

Reviewer #3:Wang et al. investigate the development of a protease-dependent regulatory mechanism of ENaC function. They specifically look into the emergence of polybasic tracts in the α and γ subunits. Furthermore, they compare transcript levels of ENaC subunits between ropefish and African clawed frog where the former independently developed lungs from tetrapods and the latter evolved after transitioning to terrestrial habitat. As it currently stands, this manuscript is focused on one topic, essentially the emergence of polybasic tract in ENaC subunits and is not well suited for a broader readership that eLife supports. There are a few comments that authors can expand on and address to create a manuscript that is more wide-reaching in terms of reader interests.1) As the authors surely know, there are other protease recognition sites that do not necessarily contain polybasic tracts. Why do ENaC subunits have polybasic tracts? Is it due to the protease furin? When did furin emerge?

We have revised the discussion our manuscript (page 13, line 3) to address this issue as follows:

“Why do ENaC subunits have polybasic tracts? Integral membrane proteins including ENaC are further processed and sorted in the Golgi apparatus for transport to their destinations. Furin and other proprotein convertases are present in various parts of the Golgi complex and cleave after polybasic motifs where Arg is preferred in the last position (22679642). Selective double cleavage strongly increases ENaC currents, providing the functional change required for fitness based selection. ENaC processing through the Golgi, where furin or other proprotein convertases are present, may have led to the emergence of polybasic tracts. Furin homologs are present in *Drosophila*, consistent with furin emerging well before ENaC subunits, and before the divergence of deuterostomes and protostomes (1915835; 16763672).”

2) Related to the first comment, are there other protease sites in any of the ENaC subunits that are not polybasic? When did they emerge?

The human ENaC γ subunit has non-polybasic protease sites adjacent to the distal polybasic site. However, the low sequence specificity of these proteases (e.g., plasmin, neutrophil elastase, and pancreatic elastase) poses a significant hurdle in tracking sites within a multiple sequence alignment. For example, neutrophil elastase has a moderate preference for cleavage after V, I, A, or T (MEROPS database). Cleavage at non-polybasic sites in non-mammalian ENaCs has also not been investigated. For these reasons, we cannot speculate as to when such sites emerged.

3) The authors provide RT-PCR data of ENaC subunits in different tissues derived from ropefish and African clawed frog. Can the authors explain the result in Figure 3 as to why there is no considerable signal of α and γ subunits, only faint bands of δ and β from frog skin? There are numerous functional data – seminal studies of ENaC, in fact – collected from frog skin that suggest a large role of ENaC in regulating Na^+^ flux. Does the subunit composition on frog skin ENaC lack α and γ subunits?

We thank the reviewer for raising this concern as it directed us to an error. RT-PCR results for *Xenopus* α and δ were previously published, and although our results were largely consistent, the results for a and d in skin were switched with respect to the previous work. For this reason and band quality issues raised by Reviewer 1, we repeated the experiment using newly harvested tissues. New data were consistent with published results for a and d (Figure 4B). We do not know the source of the original discrepancy. Nonetheless, we did not detect ENaC γ subunit transcripts by RT-PCR in any of our newer frog skin preparations. ENaC channels in frog skin may therefore lack γ subunits, or γ subunits may be present despite sub-detectable levels of transcript.